# Biogenic Synthesis and Characterization of Antioxidant and Antimicrobial Silver Nanoparticles Using Flower Extract of *Couroupita guianensis* Aubl.

**DOI:** 10.3390/ma14226854

**Published:** 2021-11-13

**Authors:** Reetika Singh, Christophe Hano, Francesco Tavanti, Bechan Sharma

**Affiliations:** 1Department of Biochemistry, University of Allahabad, Allahabad 211002, India; 2Laboratoire de Biologie des Ligneux et des Grandes Cultures (LBLGC), INRA USC1328, Université d’Orléans, Eure et Loir Campus, 21 Rue de Loigny la Bataille, 28000 Chartres, France; 3Bioactifs et Cosmétiques, Centre National de la Recherche Scientifique (CNRS)-Groupement de Recherche 3711, Université d’Orléans, CEDEX 2, 45067 Orléans, France; 4CNR-NANO Research Center S3, Via Campi 213/a, 41125 Modena, Italy

**Keywords:** nanoparticles, silver nanoparticles, antioxidant capacity, antimicrobial potential, green synthesis, *Couroupita guianensis*

## Abstract

*Couroupita guianensis* Aubl. is an important medicinal tree. This tree is rich in various phytochemicals, and is therefore used as a potent antioxidant and antibacterial agent. This plant is also used for the treatment of various diseases. Here, we have improved its medicinal usage with the biosynthesis of silver nanoparticles (AgNPs) using *Couroupita guianensis* Aubl. flower extract as a reducing and capping agent. The biosynthesis of the AgNPs reaction was carried out using 1 mM of silver nitrate and flower extract. The effect of the temperature on the biosynthesis of AgNPs was premeditated by room temperature (25 °C) and 60 °C. The continuous stirring of the reaction mixture at room temperature for approximately one hour resulted in the successful formation of AgNPs. A development of a yellowish brown color confirmed the formation of AgNPs. The efficacious development of AgNPs was confirmed by the characteristic peaks of UV–Vis, X-ray diffraction (XRD) and Fourier transform infrared (FT-IR) spectroscopy spectra. The biosynthesized AgNPs exhibited significant free radical scavenging activity through a DPPH antioxidant assay. These AgNPs also showed potent antibacterial activity against many pathogenic bacterial species. The results of molecular dynamics simulations also proved the average size of NPs and antibacterial potential of the flower extract. The observations clearly recommended that the green biosynthesized AgNPs can serve as effective antioxidants and antibacterial agents over the plant extract.

## 1. Introduction

Nanotechnology is a well-recognized and exponentially growing field with potential applications in the medical and non-medical field [1,2,3,4,5]. The wide range of applications of nanotechnology include device technology, medicine, medical imaging, drug delivery, food technology and environmental aspects [6]. Nanotechnology is a growing field that makes an impact in all provinces of human life [7], dealing with the synthesis, development and applications of a variety of nanoparticles (NPs). Usually, NPs are produced using noble metals, such as gold, silver, platinum, etc., employing a variety of physiochemical methods [8,9]; however, these methods are not eco-friendly [10,11]. The controlled biosynthesis of metal nanoparticles offers a significant effect by providing a definite size and morphology of NPs, which is different from results obtained by physicochemical methods. These NPs may be defined as particles whose sizes range from 1–100 nm. There is an urgent need to develop a sustainable method for the synthesis of NPs that does not employ noxious chemicals, in order to decrease the toxic residues released into the environment. Nowadays, green biosynthesis methods are commonly employed for NPs synthesis using various biological systems, such as yeast, fungi, bacteria and plant extract [12,13,14]. Among them, the plant-extract-based biosynthesis of silver nanoparticles (AgNPs) with controlled physicochemical properties is more popular [15,16] and has been reported with extracts obtained from several plants, such as *Iresine herbstii* [15], *Cestrum nocturnum* [16], *Carissa carandas* [17], *Mimusops elengi* [18], *Alternanthera sessilis* [19] and *Azadirachta indica* [20]. Some of the workers have used the modified methods, such as photo-irradiation-based biosynthesis and some additional agents (alginates, etc.), for the green synthesis of nanoparticles [21,22,23]. The AgNPs synthesized from biological materials exhibit an exceptionally high surface area to volume ratio with a small size, high dispersion and distinctive surface plasmon resonance (SPR) properties acting as catalysts in numerous oxidization reactions [24]. From the very beginning, the antibacterial activity of silver has been reported for many bacterial strains [25,26]. The significant antibacterial efficacy of AgNPs is a consequence of a well-expanded surface providing maximum contact with the environment [27]. The controlled size and shape, as well as the stability, of AgNPs play a significant role for their varied applications [28].

Bacterial infection is the most common cause of infectious and non-infectious diseases of human beings, such as acne and bronchitis. Antibiotics are the only way to kill the pathogenic bacteria, but the misuse and abuse of synthetic antibiotics may facilitate the bacterial species to become multi-drug-resistant [29,30]. Small molecules involving both the synthetic and plant-based natural product are the most promising agents for killing the bacteria, representing approximately 46% of the antibacterials facing preclinical trials worldwide [31]. For example, Khatoon et al. showed that both AgNPs and AuNPs exhibit antimicrobial activity against Gram-positive and Gram-negative bacteria and against fungal cells [32,33].

*Couroupita guianensis* Aubl. (family-Lecythidaceae) is popularly known as the “cannon ball tree” (Figure 1) and, in India, as “Nagalingam pushpam”, due to the shape of its flower [34]. Almost all parts of this plant, such as the leaf, flower, bark, stem and fruit’s shell, are used in the treatment of various ailments. Shivashankar et al. [35] reported the antimicrobial and antioxidant activity of the bark of *C. guianensis*. The leaves and fruit’s shell of this plant possess antibacterial activity [34,36], and the flower’s extracts have been reported for their immunomodulatory [37] and antioxidant activities [38]. The flower of this plant is rich in several phytocompounds, such as amirins, tannins, sterols, ketosteroids, etc. Tayade et al. reported the extraction of indirubin and indigo from the flower extract of *C. guainensis* [39]. Recently, Narkhede et al. showed how natural molecules present in the leaves and fruits of *C. guianensis*, such as indirubin and indigo, were able to inhibit the major protease of COVID-19; in general, they are used against coronaviruses [40,41], making these natural extracts good candidates for the development of new and more effective drugs. Earlier, Kumar et al. (2019) used the flower buds of this plant for AgNPs synthesis and Nikhitha et al. (2016) used only the petals of the flower for the biosynthesis of AgNPs [42,43], but no one used the whole flower for the extraction and biosynthesis of AgNPs. In the present study, a green biosynthesis method was used for the synthesis of AgNPs, due to its eco-friendly nature, with an innovative approach using only *C. guianensis* Aubl. flowers. This is the first report to use the green chemistry method for the synthesis of AgNPs from *C. guianensis* whole flower extracts and to explore their antioxidant and antibacterial activities. The novelty of this work resides on the usage of the whole flower extract, which contains several anti-oxidant and anti-bacterial molecules, while employing an eco-friendly process for the synthesis of AgNPs with good efficiency against Gram-negative bacteria.

## 2. Materials and Methods

### 2.1. Plant Materials and Required Chemicals

The *C. guianensis* flowers were collected from JNTBGRI campus, Palode, Trivendrum, Kerala, India. All of the chemicals used in the study were of analytical grade and purchased from Sigma-Aldrich. Millipore distilled water was used for the experiments. All of the experiments were performed in triplicates.

### 2.2. Extract Preparation from Whole Flowers

Fully blossomed mature flowers were collected from mature tree grown up to 50–60 ft. To remove the dust particles, the flowers were washed properly under running tap water and also rinsed two times by using Millipore distilled water. Flowers were dried under shade condition for 4–5 days and oven dried for 2–3 h at 40–45 °C. A mechanical grinder was used to make the coarse powder from dried flowers. A Soxhlet apparatus was used for the extraction, following same process as mentioned by Singh and Kumari [44].

A total of 100 mg of extracts was dissolved in 40 mL Millipore distilled water (MDW) and boiled for 2 min. This preparation was filtered using Whatman no. 1 filter paper and used for nanoparticle synthesis.

### 2.3. Synthesis of Silver Nanoparticles (AgNPs)

Synthesis of AgNPs was performed using green synthesis method. In brief, for the synthesis of AgNPs, 45 mL of freshly prepared aqueous solution of silver nitrate (AgNO_3_, 1 mM) was mixed with 5 mL of plant extracts (PE). The biosynthesis of AgNPs was carried out at room temperature (25 °C) and 60 °C in order to observe the effect of temperature on the synthesis of AgNPs.

### 2.4. Characterization of Silver Nanoparticles

#### 2.4.1. UV-Vis Spectroscopic Analysis

UV-Vis spectroscopic analysis of biosynthesized AgNPs was performed by continuous scanning from 350 to 700 nm (Thermo-Scientific—Evolution-201, Waltham, MA, USA) and 1 mM silver nitrate solution was used for the baseline correction.

#### 2.4.2. X-ray Diffraction (XRD) Analysis

X-ray diffraction (XRD) analysis of purified AgNPs was performed using XRD-6000 X-ray diffractometer (Shimadzu, Kyoto, Japan). XRD machine was operated at a voltage of 40 kV and a current of 30 mA, with Cu Kα radiation in θ–2θ configurations. The average size of AgNPs was calculated using the Debye–Scherrer equation by determining the width of the (1 1 1) Bragg reflection [45].

#### 2.4.3. Fourier Transform Infrared (FT-IR) Analysis of AgNPs

Formation of AgNPs was also confirmed using FTIR analysis. UV-vis and XRD analysis showed almost similar results for both preparations of nanoparticles (25 °C and 60 °C). The further analysis was performed with AgNPs synthesized at 25 °C only. AgNPs synthesized at room temperature (25 °C) were characterized using an FTIR (Bruker- Alpha, V70, Palaiseau, France) spectrometer. For the characterization, the fine powder of nanoparticles was analyzed using FTIR (in the range of 500–4000 cm^−1^) to study the biomolecules present as capping agents on silver NPs surface.

### 2.5. Antioxidant Activity of Plant Extract and AgNPs

Antioxidant activity of PE and AgNPs was calculated following the method used by Singh et al. (2021) with slight modification [17]. In brief, different concentrations of PE and AgNPs (5%, 10%, 15%, 20%, 25%, 30%, 35% and 40%) were mixed well with freshly prepared DPPH (1 mL, 0.004% in absolute methanol) solution. This reaction mixture was incubated at room temperature for 30 min in dark. DPPH solution was used as control and ascorbic acid was used as standard. The absorbance was recorded at 517 nm using UV-vis spectrophotometer (Thermo-Scientific Evolution-201). The antioxidant activity was expressed as the percentage of inhibition, which was calculated using the following formula:(1)$$\mathrm{Percentage}\;\mathrm{of}\;\mathrm{antioxidant}\;\mathrm{activity}\;=\frac{Ac-As}{Ac}\times 100$$
where *Ac* is the absorbance of control and *As* is the absorption of experimental sample.

### 2.6. Study of Antibacterial Sensitivity

The antibacterial activity of extract, AgNO_3_ and AgNPs synthesized at 25 °C was evaluated using disc diffusion method against *Morganella* spp., *Salmonella typhimurium*, *Enterobacter faecalis*, *Citrobacter* spp. and *Gonococci* spp. [46]. Pure young cultures of bacteria were obtained by sub-culturing bacterial species on MHA (Mueller-Hinton agar) medium. The agar plates were swabbed using swab stick with bacterial suspension (0.5 McFarland’s). The extract, AgNO_3_ and AgNPs solutions (5 µL of each) were dropped on sterile discs (5 mm). AgNO_3_ was used as a control. These cultured plates were incubated at 37 °C in the incubator (Heratherm Compact Thermo Scientific, Illkirch, France) for 24 h. The inhibition zone (around the disc, mm) was measured using ordinary scale.

### 2.7. Statistical Analysis

The experiments were performed in triplicate. The statistical analysis was performed using XL-stat_2018 (Addinsoft, Paris, France).

### 2.8. Molecular Dynamics Simulations(MDS)

In order to confirm experimental results, we employed computer-aided molecular dynamics simulations to obtain the atomistic structure of AgNPs and its XRD spectra profile. To build the AgNP, we employed a melt-and-quench protocol that showed good performance in the description of bulk materials [47,48]. The starting system is made up of 4000 randomly displaced Ag atoms inside a cubic box with 15 nm in size. The system is heated up to 2500 K in order to obtain a liquid system that completely loses the memory of the initial state. The liquid phase is maintained for 10 ns. Then, the system is gradually cooled down to 300 K with a cooling rate of 5 K/ps, which is a good compromise between accuracy and a quite low computational cost [48]. In the last step, the system is simulated at the constant temperature of 300 K for 10 ns in order to let the system be rearranged at ambient temperature. AgNP representations in the liquid and solid state are reported in Figure 2. This approach is different from the experimental approach, which is computationally too heavy to simulate, whereas the melt-and-quench protocol gives a realistic description of the AgNP structure and of several other systems [47,49,50]. The force field employed to simulate the AgNP formation is based on the embedded-atom model (EAM) obtained from the work of Kang et al. [50] for Ag and Cu bulk systems. This AgNP preparation had an approximate diameter of 4.5 nm with a spherical shape, which indicated that it was a good model to accurately describe the AgNP atomistic structure and its interactions with the molecules in the plant extract.

To load the AgNP with the molecules commonly found in the plant extract of *C. guianensis*, we chose five molecules whose atomistic structures are reported in Figure 2: Couroupitine A, indigo, idirubin, nerol and isatin [51]. The structure of the AgNP obtained in the first step was inserted into the center of an empty cubic box that was 9 nm in size. Then, 30 replicas of each molecule were randomly inserted into the simulation box, ensuring that they were not in contact with the AgNP. The obtained box was then filled with SPC water [52]. The system was simulated at 300 K for 100 ns while the Ag atoms positions were kept frozen, as was carried out previously [51,52,53,54]. The Gromos 54a7 force field was employed for the simulation of the adsorption of molecules over the AgNP surface [55,56,57]. The parameters for the molecules were retrieved from the Automated Topology Builder (ATB) [58] and the parameters for the AgNP were obtained from the work of Heinz et al. [59]. The combination of these parameters has already been employed with good results for Au and Ag NPs [51,52,53,54,60]. All simulations were performed with the LAMMPS package [61] and the timestep used was 0.1 fs. The temperature was controlled using a Nose–Hoover thermostat with a coupling time of 10 fs.

To simulate the interactions of the loaded AgNP with the *C. guianensis* molecules with the *S. typhimurium* bacteria, we built a simulation box containing the ST50 outer membrane protein of *S. typhimurium* and the loaded AgNP, placing it over the extracellular domain, as shown in Figure 2E. The atomistic structure of the ST50 outer membrane protein of *S. typhimurium* was retrieved from the PDB [62] (ID: 5BUN) [63]. The simulation was performed with the same parameters employed to load the AGNP with the plant extract molecules, without any bias that drives the interactions between the AgNP and the ST50 protein.

The XRD pattern was computed using the PLUMED package [64] and XRD script developed by Lodesani et al. [65].

## 3. Results

The results showed the successful formation of AgNPs from the flower extract of *C. guianensis*. The formation of AgNPs was confirmed through the following observations:

### 3.1. Silver Nanoparticles Formation

The formation of AgNPs began just after mixing the flower extract into the AgNO_3_ solution (1 mM). After the addition of extracts, the AgNO_3_ began changing its color from colorless to yellowish brown in approximately 10–15 min, and, finally, the solution turned from a light brown to dark brown color in approximately 40–45 min at room temperature. A yellowish color appeared in 5–10 min and a dark brown color appeared within 30 min at a high temperature (60 °C). Color changes were more intense at the higher temperature than at room temperature (Figure 3). The pH value of AgNPs was measured to be 7.5 and 7.1 at 25 °C and 60 °C, respectively. The results obtained from the MDS showed that the bare AgNP in the solid phase assumed a spherical shape, with an internal and surface atom geometry consisting of well-ordered structures, such as the (111), (200) and (220).

### 3.2. UV-Vis Spectroscopy Analysis

A UV–visible spectral analysis of the mixed solution (flower extract and AgNO_3_) was also performed to confirm the formation of AgNPs. The characteristic surface plasmon resonance (SPR) absorption band of biosynthesized AgNPs was obtained as 432 nm and 444 nm for the reaction carried out at 60 °C and 25 °C, respectively (Figure 3).

### 3.3. XRD Analysis

XRD analysis was also performed to observe the formation of AgNPs. The outcome of the XRD analysis of the NPs showed strong peaks corresponding to (111), (200) and (220) Bragg reflection based on the face centered cubic (fcc) structure of AgNPs (Figure 4). These planes and peaks confirmed the crystalline nature of AgNPs. These peaks (111, 200 and 220) represented the 2θ° values 38.06, 44.23 and 67.43, respectively. Scherer’s equation was used to calculate the average crystallite size, which was found to approximately be 34 ± 2 nm (25 °C) and 31 ± 3 nm (60 °C).

Figure 4D demonstrates the XRD spectrum obtained from the MDS for the liquid AgNP, solid AgNP, the molecules in the solution without the AgNP and the molecules adsorbed on the AgNP. By a visual inspection of the computer simulation results, AgNPs in the liquid phase were completely disordered, as confirmed by the XRD spectra of Figure 4D. When the system was cooled down to 25 °C, the AgNPs assumed a spherical shape while maintaining a crystalline order. It was interesting to note that the AgNPs exhibited a marked peak at 2θ = 38, which is consistent with the (111) geometry that has been visually observed from MD simulations and found by XRD experiments. This main peak was characterized by a very steep slope at 2θ = 35, corresponding to the (111) surface, and a broader shoulder at 2θ = 42, which was associated with the (200) structures, as observed in Figure 4C. The XRD spectrum obtained from the MDS was not as accurate as that experimentally obtained, due to the small size of the system being taken into account, and the broadening observed is related to the two peaks that are close to each other. Moreover, the peak at 2θ = 65 was found to correspond to the (220) surface, confirming the experimental finding that the green synthesized AgNPs assumed a crystal-like structure.

The AgNPs covered by molecules showed a similar XRD spectra with respect to the base AgNPs, but the main peak at 2θ = 38 was lower, with the shoulder at 2θ = 42 more pronounced, as experimentally observed in the FT-IR analysis of AgNPs.

FT-IR studies of AgNPs were carried out to recognize the possible biomolecules responsible for the reduction of the Ag+ ions and the capping of the bioreduced AgNPs biosynthesized by *C. guianensis* flower extracts. The FT-IR spectrum of AgNPs synthesized at 25 °C is presented in Figure 5. Various peaks of FT-IR represent different functional groups. Biosynthesized AgNPs have shown absorption peaks in regions that are already related to the presence of polyphenols capped by AgNPs [10]. The FT-IR spectra had major vibration modes at 677, 1043, 1111, 1377, 1631, 2140, 2349, 2866 and 2935. All of these spectra signified different functional groups. The results of the FT-IR analysis suggested the existence of different groups of various secondary metabolites.

### 3.4. Antioxidant Activity Studies

The results of the antioxidant assay showed a significant free radical scavenging activity of PE and AgNPs. AgNPs exhibited a higher antioxidant potential when compared to the antioxidant potential of the flower extract. The antioxidant activity of AgNPs increased from 5–20% of concentrations, whereas the antioxidant activity increased from 5–25% in the case of the flower extract. The maximum antioxidant activity (70.35%) was observed in NPs, whereas, in the case of the flower extract, the maximum antioxidant activity was 28.50% (Table 1). After 20% of the sample concentration, the antioxidant activity begins to decrease with high concentrations in AgNPs, and begins to decrease after 25% in the plant extract sample.

For comparison, the antioxidant activity acquired for AgNO_3_ (5%) was 34.81 ± 0.50%, whereas the antioxidant activity of the standard antioxidant butylated hydroxyanisole (BHA, 50 µM) was 68.12 ± 1.27%, which was slightly lower than that obtained using 20% of AgNPs.

### 3.5. Antimicrobial Activity of AgNPs

A significant bacterial growth inhibition activity was observed from the AgNPs. A higher bacterial growth inhibition was observed from the AgNPs when compared to the plant extract (PE) and AgNO_3_ (Table 2 and Figure 6). The maximum growth inhibition was observed against *S. typhimurium* (16.86 mm). Similar values of the growth inhibition zone have been observed for *Escherichia coli* (7.5 mm) and *Bacillus subtilis* (7 mm) for citrate-synthesized AgNPs [32,33].

To understand the interaction between the loaded AgNP and the *S. typhimurium* bacteria, we simulated the ST50 outer membrane protein, which was a channel protein for *S. typhimurium*. This protein contained an outer membrane (OM) domain with a β-barrel shape with three extracellular loops that open to create a flux from the periplasma to the extracellular medium [63]. We observed that these three loops fluctuated before the binding with the AgNP and exhibited a flexible structure, as shown in Figure 7. When the AgNP interacted with the ST50 OM domain, two of the three loops in contact with the NP assumed a closed conformation, and they remained in that closed position due to the binding of the AgNP, which prevented them from opening by closing the gate.

## 4. Discussion

Color changes in the reaction mixture (AgNO_3_ solution + PE) confirmed the synthesis of nanoparticles [10,17]. The color of the reaction mixture solution changed from colorless to yellowish, then to yellowish brown and finally to dark brown. The synthesis of AgNPs at room temperature takes more time when compared to the 60 °C [17], which may be due to the high temperature that speeds up the process. Some previous reports have explained that color changes in the solution showed the presence of AgNPs due to the excitation of surface plasmon vibrations [32,33,66]. A smaller size of NPs was achieved due to the rapid consumption of the reactants [67]. For the synthesis of AgNPs, the flower extracts of *C. guianensis* acted as a reductant for AgNO_3_, and, finally, AgNPs synthesis took place. The characteristic absorption spectrum from the UV-Vis spectrum analysis also confirmed the synthesis of AgNPs. The metal nanoparticles, such as silver, have free electrons, which contribute to the SPR absorption band [68]. A higher absorbance peak was obtained from AgNPs formed at 60 °C, which showed that a higher number of nanoparticles were synthesized at 60 °C. Similar UV-Vis absorption spectrum investigations have also been reported by several researchers [16,17,32,33,36] and the broadening of Bragg’s peaks also confirmed the nanoparticles synthesis. From the XRD study, both types of AgNPs (synthesized at 20 °C and 60 °C) showed an almost equal broadening of Bragg’s peaks, revealing that an almost equal size of nanoparticles is present in both solutions. Similar results of XRD spectrum patterns have been reported by many researchers [10,13,17,32,33]. The XRD peaks were very strong and indicated that the AgNPs were synthesized in a nano-regime with a crystalline nature, where the surfaces have a predominant (111) structure, as also confirmed by the MDS analysis. A similar pattern of XRD peaks was also reported when using other plants [69,70]. Kumar et al. [42] synthesized AgNPs with a size of 17 nm using the *C. guianensis* flower buds extract. Their findings suggested that the XRD spectrum contained more peaks that belonged to several crystallographic planes. This indicates that our AgNPs were more uniform from a structural point of view and suggests that smaller AgNPs synthesized with *C. guianensis* extracts have a higher degree of crystallinity with respect to bigger ones.

The FT-IR spectrum showed the presence of various functional groups and also exposed the existence of various phytochemicals in the flower extracts. A similar pattern of vibration bands has been reported in other plants [17,18,70,71]. The FT-IR spectrum band at 677, 1111 cm^−1^ represents the vibration of C=C alkenes and the presence of the methoxy group (–OCH_3_). The band at 1043 cm^−1^ was assigned for the C–N (amines) stretch vibration of the proteins. The band at 1377 cm^−1^ represents the N=O symmetry stretching, which is typical of the nitro compound. The band at 1631 cm^−1^ relates to C–N and C–C stretching, indicating the presence of proteins. The spectrum peaks of 2140 and 2349 proved the presence of alkynes N=C=O and O=C=O, respectively. The vibration bands of 2935 and 2866 have shown the presence of a –C–H– stretching of aromatic rings and the C–O group, respectively. Most of the vibration bands were similar to those in previous reports [17,18,70,71]. The results of the FT-IR study proved the presence of various phytochemical groups in *C. guianensis* flower extracts. The presence of various characteristic functional groups may be responsible for the medicinal properties of *C. guianensis.*

*C. guianensis* is an important medicinal plant that is traditionally used for the treatment of various diseases due to its antioxidant and antimicrobial activities [34,72]. Here, we decided to reconsider the traditional uses of the *C. guianensis* flower extract by using it for the green biosynthesis of AgNPs for evaluating their antioxidant and antimicrobial activities. Previous works of Khatoon et al. [32,33] showed that both Ag and Au NPs obtained from trisodium citrates as reducing agents have a good antimicrobial activity against bacteria and fungal cells. They observed that the citrate-synthesized AgNPs and AuNPs have an inhibition zone from 4 mm to 8 mm depending on the bacterial strain and on the method employed to produce the NPs [32,33], and their values are in good agreement with our results when using AgNO_3_. After the synthesis with the flower extract, a higher antioxidant activity was observed from our biosynthesized AgNPs due to the mixed potential of silver and the flower extract. A DPPH radical is typically used to calculate the presence of the antioxidants of a sample/material, expressed as its percentage of free DPPH radical scavenging activity [10]. The flower extract of *C. guianensis* is a rich source of various phytochemicals, especially the phenolics, flavonoids and alkaloids, such as indigo, indirubin and couroupitne A [73,74]. In the green biosynthesis method of AgNPs, flavonoids act as major capping agents that result in an increase in the antioxidant activity of AgNPs [10,17,75]. Our results suggest that the molecules found in the plant extract cover the AgNP surface, increasing its antibacterial activity. A high antioxidant activity of many plant-based green synthesized AgNPs has been reported by many researchers [5,16,17,19]. Our AgNPs exhibited a higher bacterial growth inhibition in comparison to the plant extract and AgNO_3_ against all of the selected bacteria. A similar bacterial growth inhibition was observed in the AgNPs of *Carissa carandas* and the plant extract of *Sapinus mukorossi* [17,76]. The antibacterial activity indicated that AgNPs biosynthesized using the *C. guianensis* flower extract were more likely to inhibit the growth of Gram-negative bacteria, especially against *S. typhi*. We observed that the antibacterial activity of AgNPs was enhanced by the molecules present in the PE that cap the AgNP surface during the synthesis process [13,23,25]. The results suggested that the antibacterial activity of AgNPs could be caused by the destruction of the microbial cell membrane [13,25]. We observed that loaded AgNPs could spontaneously bind to the ST50 protein of *S. typhi*, acting as a stopper that closes the channel. However, further studies are required to improve the antibacterial efficacy of AgNPs.

## 5. Conclusions

In this work, the *Couroupita guianensis* flower extract was successfully used for the biosynthesis of AgNPs. The formation of AgNPs was confirmed through UV-Vis spectrum analysis, FTIR analysis and XRD studies, and in conjunction with computer MDS. Green biosynthesized AgNPs have shown significant antioxidant and antibacterial activities. These AgNPs were more likely to inhibit the growth of Gram-negative bacteria, specifically against *S. typhi*. The antioxidant potential of these AgNPs can be used for scavenging several free radicals. However, an extensive research is required, especially in in vivo conditions, in order to find out the accurate dose and toxicity assessment of AgNPs before clinical practice of these nanoparticles. Moreover, future work may also be accompanied with the identification of end-capping phytocompounds responsible for these biological activities.

## Figures and Tables

**Figure 1 materials-14-06854-f001:**
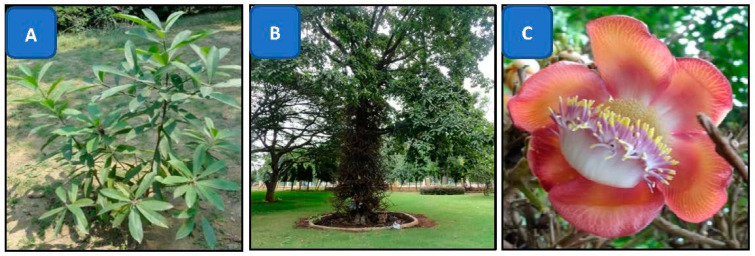
Tree of *Couroupita guianensis*: (**A**) one year old sapling; (**B**) a mature tree; (**C**) flower.

**Figure 2 materials-14-06854-f002:**
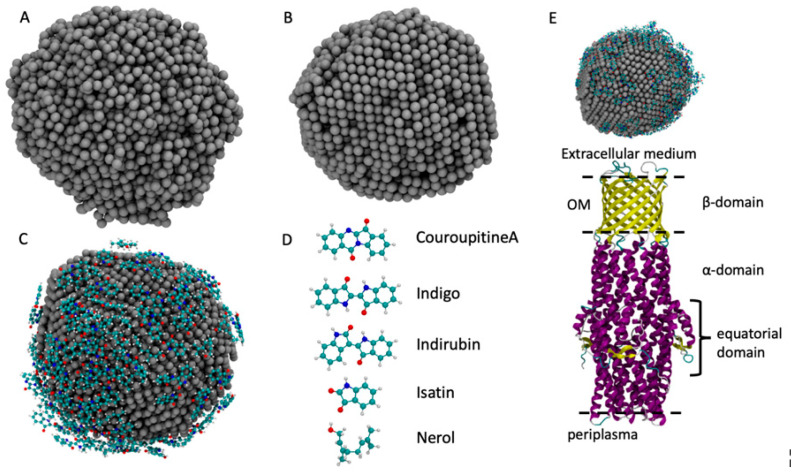
The models of the AgNP obtained from MD simulation: (**A**) the liquid phase of the AgNP at 2500 K; (**B**) the solid phase at 300 K; (**C**) the AgNP covered by molecules; (**D**) the five molecules employed in this work; (**E**) the starting configuration of the ST50 outer membrane protein of *S. typhimurrium* and of the *C. guianensis* AgNP.

**Figure 3 materials-14-06854-f003:**
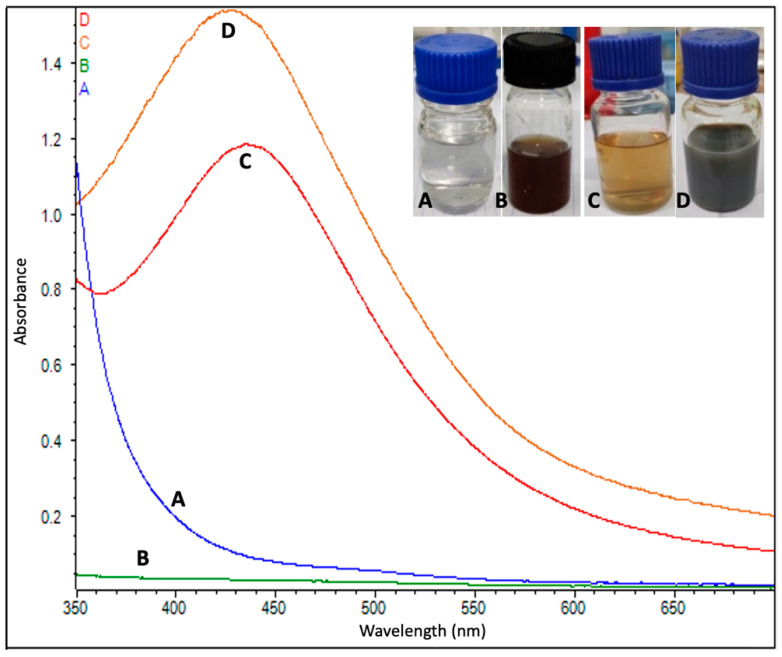
UV-vis absorption spectrum of (**A**) AgNO_3_, (**B**) leaf extract of *Couroupita guianensis* and (**C**) biosynthesized silver nanoparticles at 25 °C and (**D**) at 60 °C.

**Figure 4 materials-14-06854-f004:**
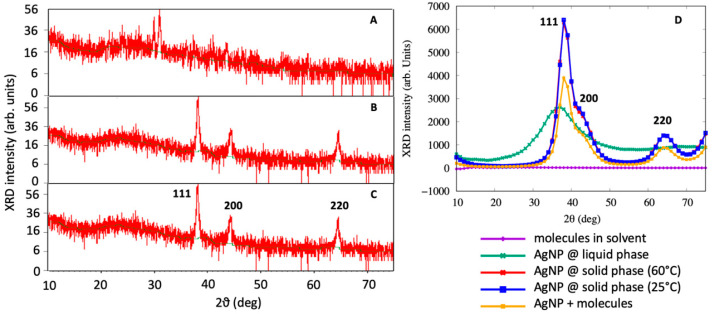
X-ray diffraction spectrum of (**A**) flower extract of *Couroupita guianensis*, (**B**) of biosynthesized AgNPs at 25 °C and (**C**) at 60 °C and (**D**) of MD simulations.

**Figure 5 materials-14-06854-f005:**
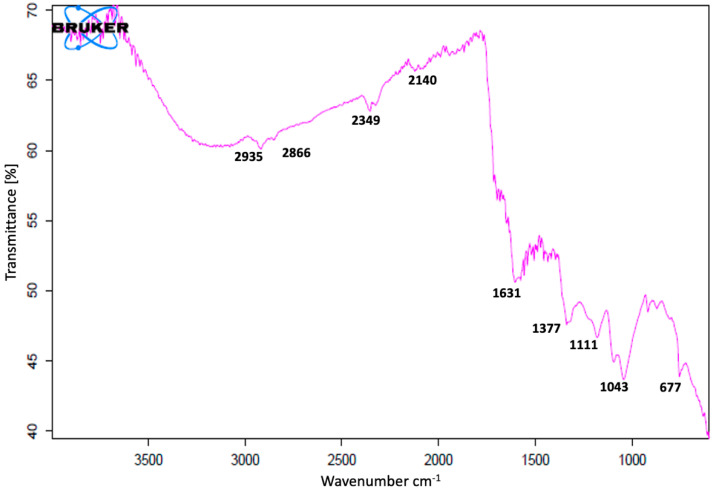
Fourier transform infrared (FT-IR) spectrum of *C. guianensis* leaf-extract-mediated biosynthesized silver nanoparticles at 25 °C.

**Figure 6 materials-14-06854-f006:**
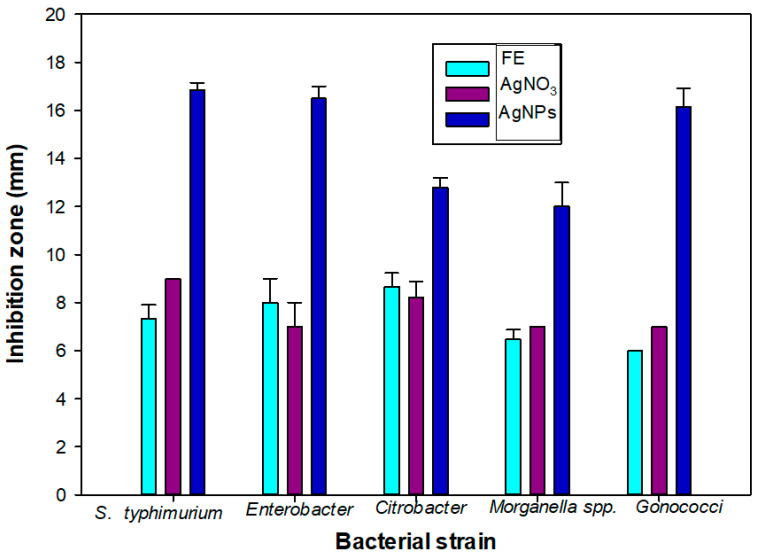
Bacterial growth inhibition zone of flower extract (FE), silver nitrate and silver nanoparticles against pathogenic bacteria (*Salmonella typhimurium*, *Enterobacter*, *Citrobacter* spp., *Morganella* spp. and *Gonococci*).

**Figure 7 materials-14-06854-f007:**
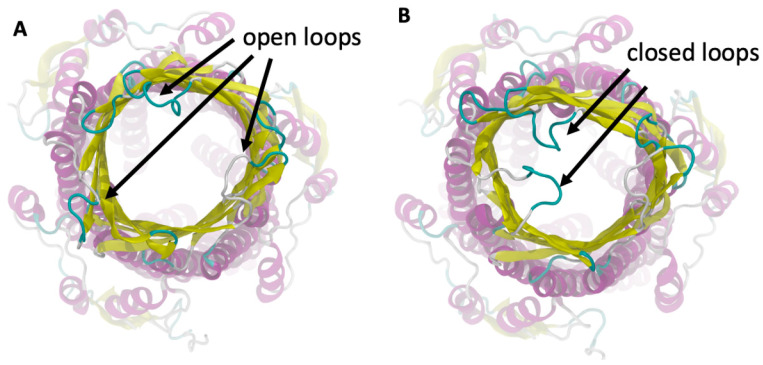
Outer membrane domain of the ST50 protein in the open (**A**) and closed (**B**) position.

**Table 1 materials-14-06854-t001:** Antioxidant Activity of Plant Extract and AgNPs.

Concentration (%)	Antioxidant Activity Flower Extract	AgNPs (25 °C)
5	15.32 ± 0.17 h	50.53 ± 0.59 f
10	20.24 ± 0.24 f	65.16 ± 0.10 c
15	23.55 ± 0.17 cd	68.63 ± 0.16 b
20	25.40 ± 0.45 b	70.35 ± 0.11 a
25	28.50 ± 0.30 a	67.47 ± 0.25 b
30	24.65 ± 0.31 cd	64.43 ± 0.25 c
35	22.40 ± 0.36 e	61.96 ± 0.24 de
40	19.39 ± 0.12 g	60.46 ± 0.23 de

Values are averages with standard deviations of three independent data sets. Different letters represent significant differences between the different antioxidant activity values (*p* < 0.05).

**Table 2 materials-14-06854-t002:** Bacterial Growth Inhibition Potential of Biosynthesized AgNPs from *C. guianensis* Flower Extract.

Bacteria	Growth Inhibition Zone (mm)		
	Flower Extract	AgNO_3_	AgNPs
*S. typhimurium*	7.33 ± 0.57	9.00 ± 0.00	16.86 ± 0.30
*Enterobacter*	8.00 ± 1.00	7.00 ± 1.00	16.5 ± 0.50
*Citrobacter*	8.66 ± 0.57	8.23 ± 0.66	12.8 ± 0.40
*Morganella* spp.	6.46 ± 0.41	7.00 ± 0.00	12.00 ± 1.00
*Gonococci*	6.00 ± 0.00	7.00 ± 0.00	16.16 ± 0.76

Values are averages with standard deviations of three independent data sets. Ciprofloxacin was used as a standard antibiotic, showing an inhibition zone of 26.55 ± 0.25 mm.

## Data Availability

The data presented in this study are available on request from the corresponding author.

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
