# Peer review of "Biogenic Synthesis and Characterization of Antioxidant and Antimicrobial Silver Nanoparticles Using Flower Extract of Couroupita guianensis Aubl."

_materials, 2021, doi:10.3390/ma14226854_

Round 1
Reviewer 1 Report
The work entitled “Biogenic synthesis and characterization of antioxidant and antimicrobial silver nanoparticles using flower extract of Couroupita guianensis Aubl.”by Singh et al. introduced the extract C. guianensis as a precursor for AgNPs synthesis. Even though the work has some merit by using a specific flower with antimicrobial and antioxidant potential, I fail to see the novelty. The methodology is the same employed by many others researching other extracts. The presentation of the data is extremely rudimentary, the authors paid little attention to the form they presented their graphics and even the pictures they took have low quality to support their conclusions. Further, the discussion is mostly the repetition of the results presented and there is barely criticism or support from the literature. The English writing is very deficient making it somewhat difficult in some instances to understand what the authors would like to convey.
Overall, I do not see the quality of this manuscript to fit the demands and standards of this journal. I recommend a thorough revision and a resubmission.
Author Response
- The work entitled “Biogenic synthesis and characterization of antioxidant and antimicrobial silver nanoparticles using flower extract of Couroupita guianensis Aubl.”by Singh et al. introduced the extract C. guianensis as a precursor for AgNPs synthesis. Even though the work has some merit by using a specific flower with antimicrobial and antioxidant potential, I fail to see the novelty. The methodology is the same employed by many others researching other extracts.
Ans. I have used the Soxhlet apparatus prepared extract for the synthesis of AgNPs whereas other authors had used the extracts by boiling of flowers in the water. We have included the MD simulation study to include the novelty in this paper and to support experimental results.
- The presentation of the data is extremely rudimentary, the authors paid little attention to the form they presented their graphics and even the pictures they took have low quality to support their conclusions.
Ans. We have improved the data presentation and graphs. We have also changed the photos of the plants.
- Further, the discussion is mostly the repetition of the results presented and there is barely criticism or support from the literature. The English writing is very deficient making it somewhat difficult in some instances to understand what the authors would like to convey. Overall, I do not see the quality of this manuscript to fit the demands and standards of this journal. I recommend a thorough revision and a resubmission.
Ans. We have improved the results and discussion part as well as the English of the manuscript. We have included the MD simulation study to improve the quality of the paper and to support the results of the previous experiments. At the best of our knowledge, this is the first time that the interaction of silver nanoparticles covered by molecules from the plant extract of C. guianensis with Typhi outer membrane protein is studied by means of computer simulations. Moreover, we modified the English style, accordingly to referee’s comments.

Reviewer 2 Report
This paper proposes a method for biogenic synthesis of silver nanoparticles with flower extract of C. guianensis Aubl. The work of this paper is clear and logical. However, I have to reject it because of the following problem:
1. This paper is not innovative enough. Authors need to highlight their innovative contributions. As the authors said, the antibacterial activity and antioxidant activity of the plant have been reported. And they announced this was the first report to use the green chemistry method for the synthesis of AgNPs from C. guianensis flower. However, the work of T. Venkata Rajesh Kumar and K. P. Nikhitha had been published in 2016 and 2019, respectively.
(T. Venkata Rajesh Kumar, J. S. R. Murthy, Madamsetti Narayana Rao, Y. Bhargava. Evaluation of silver nanoparticles synthetic potential of Couroupita guianensis Aubl., flower buds extract and their synergistic antibacterial activity. Biotech, 2016, 6:92.
P. Nikhitha, H. T. Navyashree. Green synthesis of silver nanoparticles from Couroupita guianensis aubl. flower petal extract and its antimicrobial activity. International Educational Applied Research Journal, 2019, 3(8):59-62.)
2. Too few experiments to support the results, such as the size of silver nanoparticles. The authors calculated the size from the XRD Diffraction spectra, but the signals of AgNPs at 60 ℃ were fuzzy. The result needs more demonstration, like TEM or SEM images.
Reviewer 3 Report
My comments are below;
- The aim of the manuscript very vague, if biosynthesis is suggested, there are various studies with such aims. The authors need to add novelty to their work.
- I urge authors to add more literature review, it will be helpful for the authors. Followings can be added to the antimicrobial activities.
- Fabrication of Alginate Fibers Loaded with Silver Nanoparticles Biosynthesized via Dolcetto Grape Leaves (Vitis vinifera cv.): Morphological, Antimicrobial Characterization and In Vitro Release Studies
- Biosynthesis of silver nanoparticles by bamboo leaves extract and their antimicrobial activity
- Photo-irradiation based biosynthesis of silver nanoparticles by using an ever green shrub and its antibacterial study
- The results are not presented well, the XRD curves has too much noise, should ne de-noised.
- Why FTIR has one curve with one temperatures extract?
- The antioxidant activities are confusing, read more articles to add the uses of AgNPs for antioxidant activities.
Round 2
Reviewer 1 Report
The authors have implemented significant alterations in their manuscript. Even though I still believe this work has little novelty I can see the merit of the authors arguments and after the improvements they introduced in their manuscript can now recommend its publication.
Reviewer 3 Report
The manuscript is in much better form than before, however, far from publication. The result presentation must be improved, the figures are low quality most of the axis and legends are not readable. The authors must add novelty, the claim of no other studies using such extract is low, there are many studies with similar plants. These are some of them;
Evaluation of silver nanoparticles synthetic potential of Couroupita guianensis Aubl., flower buds extract and their synergistic antibacterial activity
Novel Silver Nanoparticles Synthesized from Anthers of Couroupita Guianensis Abul. Control Growth and Biofilm Formation in Human Pathogenic Bacteria
I suggest the authors to revisit the purpose of this study, bring novelty and rewrite whole manuscript accordingly.
